# Structural Chromosome Instability: Types, Origins, Consequences, and Therapeutic Opportunities

**DOI:** 10.3390/cancers13123056

**Published:** 2021-06-19

**Authors:** Sebastián Omar Siri, Julieta Martino, Vanesa Gottifredi

**Affiliations:** 1Cell Cycle and Genome Stability Laboratory, Fundación Instituto Leloir, C1405 BWE Buenos Aires, Argentina; vgottifredi@leloir.org.ar; 2Instituto de Investigaciones Bioquímicas de Buenos Aires (IIBBA), Consejo Nacional de Investigaciones Científicas y Técnicas (CONICET), C1405 BWE Buenos Aires, Argentina

**Keywords:** chromosome instability, chromosome aberrations, chromosome bridges, lagging chromosomes, ultra-fine bridges, micronuclei, DNA damage, DNA repair

## Abstract

**Simple Summary:**

Chromosome instability (CIN) is characterized by an increased accumulation of numerical and structural changes in chromosomes and is a common feature of solid tumors and some hematological malignancies. CIN has been extensively linked to tumorigenesis, cancer progression, and tumor resistance. However, in recent years CIN phenotypes are increasingly being harnessed for therapeutic purposes. In this review, we describe the origins of structural CIN phenotypes and highlight novel pathways for their resolution. We also discuss how CIN can be avoided or enhanced and the implications of these pathways for cell survival and thus, cancer treatments.

**Abstract:**

Chromosomal instability (CIN) refers to an increased rate of acquisition of numerical and structural changes in chromosomes and is considered an enabling characteristic of tumors. Given its role as a facilitator of genomic changes, CIN is increasingly being considered as a possible therapeutic target, raising the question of which variables may convert CIN into an ally instead of an enemy during cancer treatment. This review discusses the origins of structural chromosome abnormalities and the cellular mechanisms that prevent and resolve them, as well as how different CIN phenotypes relate to each other. We discuss the possible fates of cells containing structural CIN, focusing on how a few cell duplication cycles suffice to induce profound CIN-mediated genome alterations. Because such alterations can promote tumor adaptation to treatment, we discuss currently proposed strategies to either avoid CIN or enhance CIN to a level that is no longer compatible with cell survival.

## 1. Introduction

Genomic stability is conserved by the coordinated activity of multiple genome maintenance pathways that ensure faithful DNA replication and equal distribution of duplicated DNA among daughter cells [1]. These pathways include cell cycle checkpoints, DNA damage detection and its repair, telomere maintenance, and centrosome duplication, to name a few. Their defects lead to genomic instability, a state with an increased tendency to acquire genetic alterations. Genomic instability is considered both an early step of tumorigenesis and an enabling characteristic of tumors, propelling tumor heterogeneity and providing growth advantages [2]. Mutations in genome maintenance effectors and regulators are not only sporadic; they also give rise to various genetic syndromes characterized by cancer predisposition and early cancer onset [3].

The genetic changes associated with genomic instability range from single nucleotide loss/modification to elimination or gains of whole chromosomes. Chromosome instability (CIN) is a subtype of genomic instability, which refers to an increased rate of acquisition of gross numerical or structural changes in chromosomes. CIN is a feature that is remarkably frequent in most solid tumors and some hematological malignancies and is associated with resistance [4]. Intriguingly, however, in certain tumor types (e.g., breast, thyroid, and colon cancer), CIN is often considered as a possible therapeutic target, raising the question of which variables may convert CIN into an ally during cancer treatment [5]. Thus, it is critical to unravel the triggers and consequences of each type of CIN. The easy visualization of morphological changes in chromosomes, which we will discuss in this review, should facilitate such a goal. Nevertheless, our understanding of CIN from the perspective of its molecular triggers, its role in different tumor contexts, and the procedures to prevent or exacerbate CIN remains very limited and deserves immediate attention.

## 2. Types of Chromosome Instability (CIN)

CIN refers to an increase in the acquisition rate of chromosome alterations and is classified into two types: numerical CIN and structural CIN. Numerical CIN refers to the loss or gain of entire chromosomes and leads to aneuploidy, which is distinct from the gain of a complete set of chromosomes, referred to as polyploidy. It is thought that a polyploid state (usually tetraploidy) followed by chromosome gain and loss is a key variable for building the complex karyotypes of solid tumors [6]. On the other hand, structural CIN refers to structural changes in chromosomes, including gross chromosome rearrangements, such as amplifications and deletions of parts of chromosomes, and translocations between non-homologous chromosomes [7]. An important distinction is that chromosome abnormalities may also exist without giving rise to unstable genomes (i.e., CIN), as is the case, for example, for Down syndrome patients, which are aneuploid for chromosome 21 (trisomy).

Numerical CIN is usually associated with chromosome segregation errors during mitosis, resulting from aberrant mitotic spindles, sister chromatid cohesion defects, improper microtubule-kinetochore attachments, chromosome condensation defects, or abnormal cytokinesis [8]. On the other hand, structural CIN is commonly associated with replication stress (stalled or collapsed replication forks), telomere dysfunction, and errors in the repair of double-strand breaks (DSBs) [9,10]. Despite their apparently distinct origins, it is worth mentioning that both numerical and structural CIN tend to coexist in tumors, and each can be the source of the other [10,11].

## 3. Cellular Phenotypes Associated with CIN

In cells, CIN manifests as chromosomal phenotypes that can be observed via FISH, immunofluorescence, and staining of metaphases, among other methods. These chromosomal phenotypes are best visualized during mitosis when chromosome condensation is at its peak, and it is generally accepted that their quantification is a good predictor of CIN levels. Their manifestation is also used to infer the roles of proteins involved in genome maintenance.

### 3.1. Chromosome Aberrations

Chromosome aberrations include structural changes such as large translocations, deletions, amplifications, chromosome gaps, chromosome breaks, and radial chromosomes, which can be spotted in stained metaphase spreads. While gaps may reflect single-stranded DNA (ssDNA) regions, breaks and radial chromosomes derive from the aberrant or incomplete processing of DSBs [12] (Figure 1). DSBs can be induced by DNA damaging agents or occur spontaneously, albeit at a low frequency. Breaks and radial chromosomes abruptly accumulate in cells incapable of repairing DSBs by homologous recombination (HR) or in cells that cannot repair inter-strand crosslinks [13]. Radial chromosomes are formed after the aberrant processing of DSBs by non-homologous end joining (NHEJ) [12]. On the other hand, breaks may represent unresolved DSBs or be triggered by aberrant resolution of HR intermediates [14]. Given that these aberrations and their underlying mechanisms have been thoroughly reviewed in the past, they will not be a subject of this report [9].

### 3.2. Chromosome Bridges

Chromosome bridges are DNA bridges that form when a chromosome region is simultaneously pulled to both poles of the mitotic spindle during chromosome segregation (Figure 1). There are two types of chromosome bridges that are classified based on their ability to be detected or not after staining with DNA intercalating dyes. 

#### 3.2.1. Bulky Chromosome Bridges

Bulky chromosome bridges are visible as DAPI-positive (and other DNA intercalating dyes) DNA tracks, implying that the bridged DNA is double-stranded and chromatinized. They are detectable during the anaphase and cytokinesis phases of mitosis. Remarkably, their processing is considered a source of rapid and aberrant reorganization of the genome [15]. The origins of bulky chromosome bridges are diverse. One source is the loss of the end-capping shelterin complex of telomeres that leads to unprotected DNA ends, which are mistakenly ligated together by DSB repair mechanisms such as NHEJ [16]. When non-homologous chromosomes are joined together, they become dicentric, and when pulled to opposite poles during mitosis, they give rise to bulky chromosome bridges [17].

Replication-associated one-ended DSBs (Figure 1) are another source of these bridges. Such DSBs, formed after replication fork stalling and nuclease-mediated processing of a stalled fork, can form bulky chromosome bridges due to aberrant processing by HR [18,19,20,21]. HR-mediated repair of one-ended DSBs involves the formation of X-shaped DNA structures known as Holliday junctions (HJs), which must be adequately dissolved or resolved to ensure accurate repair. A failure to execute either HJ dissolution or resolution can give rise to bulky chromosome bridges [22]. Bridges can also form without any DNA damage, as has been observed during prolonged mitotic arrest in which deregulation of separase leads to incomplete removal of cohesin from chromosome arms and chromosome non-disjunction [23].

#### 3.2.2. Ultra-Fine Bridges (UFBs)

UFBs are another type of DNA bridge that differs from bulky chromosome bridges in their origin and their inability to be stained with conventional DNA dyes due to lack of histones [24]. However, they can be visualized because of their co-localization with the Fanconi anemia protein, FANCD2, the PICH translocase, as well as by components of the BTRR complex such as the BLM helicase [25,26]. UFBs can be further processed in the following replication cycles. In contrast to bulky chromosome bridges, they may be involved in a pathway that promotes, rather than restrains, genomic stability [27]. UFBs are formed as a consequence of the accumulation of under-replicated DNA (UR-DNA) (Figure 1). Incomplete DNA replication gives rise to UR-DNA regions trapped between two stalled forks with no internal dormant origin that can fire to complete replication. Because the S to G2 transition is triggered seemingly by the absence of replication and not by the finalization of DNA replication [28], cells enter mitosis with such UR-DNA regions. If they are still not duplicated by the end of M phase, such UR-DNA regions can lead to non-disjunction (i.e., failure of chromosomes to separate in mitosis), giving rise to UFBs [29,30,31]. 

Intriguingly, UR-DNA accumulation at the end of S phase is potentially frequent in cells with large genomes. Such a phenomenon is potentiated by the accumulation of DNA replication barriers or other conditions such as nucleotide depletion [27]. There are DNA regions poor in replication origins that are more prone to under-replication, such as common fragile sites. Such regions are rich in A-T and are, therefore, predisposed to forming secondary DNA structures that halt the replication machinery. Besides increasing the chances of collisions between the replication and transcription machineries, common fragile sites code for long genes [32]. The increased frequency of formation of replication barriers, combined with the poorness in replication origins of these DNA regions, increases the chances of double fork stalls [33,34].

Other DNA regions prone to under-replication and generation of UFBs include centromeric and telomeric regions. Similarly to common fragile sites, centromeric regions replicate late and are also prone to forming secondary DNA structures, while telomeric regions contain repetitive DNA sequences that can stall or slow the replication machinery. Lastly, ribosomal loci can also give rise to UFBs. The need to maintain a constant supply of ribosomal RNA leads to the formation of persistent DNA-RNA hybrids, which cause replication defects and ultimately result in UR-DNA. UFBs can also arise from double-stranded DNA catenanes or unresolved HR intermediates [25,35,36,37]. 

### 3.3. Lagging Chromosomes 

Lagging chromosomes or laggards are chromosomes that lag in the metaphase plate during anaphase (Figure 1). Similarly to bulky chromosome bridges, they can be visualized by DAPI staining and revealed in cells transiting anaphase. They are mainly associated with defects intrinsic to mitosis such as abnormal microtubule-kinetochore attachments, centrosome amplification, aberrant spindle assembly checkpoint, and defects in sister chromatid cohesion. One of the most frequent mechanisms involved in lagging chromosome formation is the presence of merotelic microtubule-kinetochore attachments, in which one kinetochore is attached to both poles of the mitotic spindle [38]. In tumor cells, another source of lagging chromosomes is the hyper-stabilization of microtubule-kinetochore attachments, which renders cells unable to correct aberrant attachments, a vital aspect of proper mitosis [38,39]. Once formed, lagging chromosomes that are not reabsorbed into daughter nuclei can be lost in future cell divisions or form MN, leading to numerical or structural CIN [39,40,41]. 

Additionally, DNA replication stress can also lead to lagging chromosomes. Recent work shows that mild replication stress causes mis-segregation due to multipolar mitotic spindles formed by premature centriole splitting [42]. Replication stress can also increase microtubule stability, which not only contributes to the premature centriole splitting but can also have a profound impact on the ability of cells to correct aberrant microtubule-kinetochore attachments [10,42,43]. Moreover, it has also been proposed that unresolved replication intermediates that give rise to broken chromosomes and bridges cause perturbations of the spindle, facilitating the formation of laggards [36].

### 3.4. Micronuclei 

Micronuclei (MN) are small peri-nuclear bodies formed by a nuclear-membrane-like envelope that contains chromosome fragments or whole chromosomes (Figure 1). MN are easily identified after DAPI staining and accumulate during cancer genesis and treatment but can also arise due to changes in cellular metabolism during senescence, aging, and the onset of different diseases [44,45]. MNs are considered markers of genotoxic events and CIN [46], and their presence is very common in most solid tumors, neoplastic lesions, and peripheral lymphocytes of patients that develop cancer [47,48]. Interestingly, although the morphological markers mentioned above (UFBs, bulky chromosome bridges, and acentric fragments) have different origins and resolution mechanisms (see Section 6 below), it seems that all of these aberrations sooner or later may end in the formation of MN, which may explain the widespread presence of MN in tumors and also highlights the tight association among different CIN phenotypes (Figure 1).

As previously stated, MN largely derive from existing aberrations (Figure 1). Mechanistically, one source of MN is acentric chromosomes or acentric chromosome fragments that lack a centromere and thus, fail to segregate to daughter nuclei (Figure 1). MN can also be generated during aberrant DSB repair [49,50], which gives rise to dicentric chromosomes (i.e., chromosomes with two centromeres), leading to chromosome bridges during the late stages of mitosis. Such chromosome bridges can be broken by nucleases or mechanical forces from the cytoskeleton, leading to MN formation (Figure 1) [15,17,51]. Several other aberrant events lead to MN formation. For example, MN can also be formed by aggregates of double minutes, which are extrachromosomal bodies composed of circular DNA that lack centromeres and telomeres [52]. Another origin for MN includes broken, aberrantly, or incompletely processed UFBs [53], whose multiple origins were discussed above (Figure 1). Given that common fragile sites are regions prone to accumulate UR-DNA and thus form UFBs, they are also considered a source of MN in cells [34]. In subsequent cell cycles, MN can be further incorporated in the genome or get excluded from the cell and lost (see Section 6 below).

## 4. The Role of the DNA Damage Response in the Prevention of CIN

Cells have an evolutionarily conserved set of DNA damage response (DDR) mechanisms, in which different proteins are responsible for detecting DNA damage and triggering multiple cellular responses, including DNA repair [54,55]. The proper and timely activation of DDR mechanisms, such as DNA repair and DNA damage tolerance, is crucial to prevent CIN generated after replication stress. Even in optimal DNA repair conditions, the chances of replication forks running into DNA damage and other replication barriers are high, posing challenges to completing DNA replication [27]. Because replicative DNA polymerases are incapable of using damaged DNA bases as templates for DNA replication, they stall when encountering damaged DNA. At this stage, the continuity of DNA replication can be granted by DNA damage tolerance mechanisms that revert the stalling and favor DNA replication leaving the damaged DNA behind the ongoing fork. One major DNA damage tolerance pathway, translesion DNA synthesis, is driven by specialized DNA polymerases that are recruited to the damaged site and can replicate across a lesion [56,57]. 

Another DDR mechanism that facilitates the replication of damaged DNA is the intra-S phase cell cycle checkpoint. While this checkpoint does not favor DNA damage tolerance, it generates a soluble signal (active CHK1 kinase) that orchestrates DNA replication choreography in a manner that reduces the chances of replication stress [58,59,60]. If DNA damage tolerance events or checkpoint signaling are not activated, or are inefficient or insufficient, an increasing number of replication forks may not be able to cope with damaged DNA, leading to the stalling and eventual collapse of replication forks. The latter events may trigger the removal of replisome components from the fork and, subsequently, its breakage into one-ended DSBs. Once collapsed, the proper duplication of such DSBs depends on replication-coupled DNA repair mechanisms such as HR. HR participation in the repair of stalled or collapsed forks has been extensively reviewed elsewhere [61]. Despite the availability of repair mechanisms during replication, on occasion they may not be sufficient or may fail, generating undesirable processing of DSBs by error-prone pathways, thus leading to morphological rearrangements such as insertions, deletions, and fusions, or numerical abnormalities [19]. 

## 5. The Crucial Role of Nucleases in CIN Prevention 

Among the leading players in the repair of replicative DNA damage are nucleases. Nucleases are crucial both for DSB formation and for their repair. Persistently stalled forks need to be converted into one-ended DSBs by endonucleases such as MUS81 and EXO1 [62]. Exonucleases then play a central role in the generation of HR-proficient DNA ends [63]. Then, nuclease-independent events drive HR-mediated DNA synthesis with HJ resolution as the final repair step. HJs are primarily resolved by helicase-mediated (BTRR complex) resolution [64], but nucleases can also resolve HJ. The dissolution of HJs is mediated by the structure-specific endonucleases MUS81, SLX1-SLX4 [65], and GEN1 [66], which cleave DNA structures that would otherwise compromise proper chromosome segregation [36,67]. By resolving one-ended DSBs, endonucleases prevent DNA replication intermediates from forming bulky chromosome bridges, UFBs, and micronuclei, among others [30,65,68,69]. Thus, proper endonuclease activity suppresses genomic instability associated with replication fork stalling or collapse. Endonucleases also play a crucial role during mitotic DNA synthesis (MiDAS), a DNA synthesis event that also requires nuclease-mediated processing of UR-DNA and serves as the last resource available to prevent UFB formation [70,71]. However, under certain conditions, the same endonucleases can promote CIN instead of preventing it. For example, in CHK1 depleted samples, MUS81 and its cofactor EME1 augment bulky chromosome bridges and MN frequency in a manner that is associated with nucleoside-shortage mediated aberrant processing of UR-DNA [72,73].

## 6. Resolution of Structural CIN Phenotypes 

As mentioned above, replication stress is a common source of DNA damage, and cells have multiple mechanisms to overcome this burden. When these mechanisms fail, replication stress gives rise to structural CIN features such as bulky chromosome bridges, UFBs, lagging chromosomes, broken and radial chromosomes, acentric fragments, and MN. In recent years, we have begun to understand that cells also have different options to resolve these complex aberrations. Such structural CIN-processing mechanisms, which can induce different ranges of genomic instability, will be described below. 

### 6.1. Chromothripsis

Chromothripsis is a pathological resolution of both bulky chromosome bridges and MN. It refers to the rearrangement of hundreds of small chromosome fragments in a single stochastic event [74,75]. MN lack a proper nuclear envelope and are subject to the action of cytoplasmic nucleases such as 3-prime repair exonuclease 1 (TREX1), which can shatter the MN into fragments that are later re-organized, and re-incorporated into the genome [76]. MN can also undergo asynchronous replication in the following cell cycle, giving rise to unresolved replication intermediates and, thus, DSBs that are randomly joined by repair mechanisms following mitotic entrance [74,77]. This phenomenon has also been observed when bulky chromosome bridges break, and their fragments are subject to local rearrangements or “local jumps”. To generate DSBs, bulky anaphase bridges can also be cleaved by TREX1 that comes into contact with DNA due to improper nuclear membrane formation around the bridges [17]. A TREX1-independent model has also recently been proposed in which chromothripsis of bulky bridges is due to a combination of actin-dependent mechanical forces (stretching and contractile) and myosin II motor protein activity [15,78,79]. 

Multiple studies associate chromothripsis events with alterations in DNA damage repair pathways such as NHEJ, and at the same time, chromothripsis can also induce these alterations. As such, chromothripsis can be a cause or consequence of failed repair mechanisms [80] and is tightly associated with cancer progression [81]. Because chromothripsis-derived breaks can form new MN, it could efficiently propel massive genome changes and tumor heterogeneity within a minimal number of cell cycles [15]. Supporting such a notion, it was recently shown that MN-derived extrachromosomal DNA subject to chromothripsis produces double-minutes with gene amplifications. Upon formation of DNA breaks, such as those formed in chemotherapy treatments, the double minute fragments can re-integrate to chromosome ends, thus forming chromosome bridges that suffer from break-fusion-bridge cycles and further chromothripsis, leading up to more gene amplification and genomic instability [82]. Furthermore, while cell death can be avoided when resolving bulky chromosome bridges and MNs using chromothripsis, the alteration of genomic sequences is an inevitable scar that remains in the daughter cells even when the original chromosome alteration is no longer present.

### 6.2. Tethering of Broken Chromosomes

As mentioned earlier, unrepaired DSBs give rise to acentric fragments, which form MN [51,83], triggering chromothripsis and genomic rearrangements, which may no longer be visualized as structural CIN. However, live imaging studies in several systems such as Drosophila, plant, and yeast cells show that acentric fragments do not necessarily generate MN. Instead, three distinct mechanisms capable of driving segregation and successful poleward transport of acentric fragments have been described (i.e., kinetochore-independent lateral attachments to microtubules, direct attachments to other chromosomes, and long-range tether-based attachments). The long-range tether-based attachments connect the broken ends of the centric and acentric fragments allowing reabsorption of the acentric fragments into the daughter nuclei via a nuclear pore-mediated transport [40,84,85,86,87]. 

Because they are not attached to the spindle, acentric fragments may delay nuclear membrane assembly, regulated by the mitotic kinase Aurora B. During anaphase, Aurora B is removed from chromosomes and relocalized to the mitotic spindle midzone, where acentric fragments also congregate. Finally, Aurora B removal from chromosomes allows for the dephosphorylation of H3(S10), leading to the recruitment of HP1α, a protein responsible for reestablishing the nuclear envelope. In contrast, Aurora B remains enriched at chromatin in acentric fragments facilitating persistent H3(S10) phosphorylation, which prevents HP1α-dependent nuclear membrane reassembly around the acentric fragments [88]. The delayed nuclear envelope assembly may facilitate the resolution of this type of structural CIN as acentric fragments are not separated from their centric chromosome. They remain tethered by thin DNA threads reminiscent of unresolved DNA replication intermediates and contain mitotic proteins, including Polo kinase, Aurora B, INCENP, and BubR1. This tethering, which may be far more widespread than previously considered, in combination with the delay in nuclear membrane reassembly around them, forms “channels” through which these fragments can return to daughter nuclei via a BubR1 and Polo-like kinase 1-dependent mechanism [84,87,88,89,90]. According to this model, the chances are that broken chromosomes that remain attached to the nucleus by DNA threads may avoid genomic instability when structural CIN is resolved. However, this hypothesis remains to be tested.

A recent study shows that mammalian cells can also use “chromosome tethering” between DSBs that cannot be repaired in mitosis. During mitosis, the scaffold MDC1 recruits the DDR mediator TOPBP1 to sites of DSBs. The recruitment of TOPBP1 and its interaction with MDC1 serve to form filamentous assemblies that bridge MDC1 foci during mitosis and hold the DSB together until it can be repaired in G1, thus preventing acentric fragments and micronuclei. In this way, chromosome tethering can be used as a compensatory mechanism to deal with mitotic DSBs until they can be repaired [91].

### 6.3. Reincorporation of Whole Chromosome-Derived Micronuclei to the Main Nucleus

Live imaging has revealed that lagging chromosomes can, on some occasions, be distributed to the right side of the spindle, thus giving rise to whole chromosome-containing MN. These MNs are independent bodies and, during the following cell divisions, can be re-incorporated into the main nucleus [38,40,92,93]. During cell division and upon breakage of nuclear membranes (of both the main nucleus and MN), the nucleus loses track of the MN. The non-nuclear origin of MN DNA is not relevant at that point, and the spindle can separate all the chromosomes into two daughter nuclei regardless of their source. One necessary requisite is that MN DNA must duplicate correctly and fully during S phase. The improper nuclear pore assembly of MN and the challenging recruitment of replication factors to the MN’s inner side may impose some challenges to such a replication endeavor [94]. Despite such limitations, these studies collectively suggest those whole chromosomes that end up being encapsulated in a MN have a chance to return to the main nucleus, thus avoiding CIN and potential genomic instability altogether. 

### 6.4. Resolution of Ultra-Fine Bridges (UFBs)

In contrast to bulky chromosome bridges, no reports indicate that UFBs undergo mechanical breakage. Instead, the BLM helicase and PICH translocase recognize and bind to UFBs, where they recruit factors involved in UFB resolution [25,95]. Such factors may differ depending on the origin of the UFBs. For example, to resolve UFBs derived from double-stranded DNA catenanes, TOP2A is the preferred candidate, while UFBs derived from unresolved replication intermediates can be resolved via TOP3A cleavage with aid from TOP1, which separates the ssDNA hemi-catenanes that hold DNA together [96,97]. Additionally, these steps can be complemented by a previous excision by structure-specific endonucleases [98]. Lastly, UFBs that are formed by sister chromatid junctions due to unfinished HR can be resolved either by BTRR complex-mediated dissolution or by SLX1-4, MUS81-EME1, and XPF-ERCC1 or GEN1 mediated-resolution [14,36,62,98]. It is unclear if the resolution of UFBs by any of these methods prevents alterations of the DNA sequence initially compromised during UFB formation. It must also be highlighted that in all cases, these types of resolutions imply the inheritance of ssDNA regions to daughter cells. Such under-replicated ssDNA regions are shielded in 53BP1 bodies during the G1 phase (described below) and duplicated in the next S phase.

As already described above, UR-DNA regions that enter mitosis, and whose replication is not finished by MiDAS, give rise to UFBs. Before their separation between daughter cells, the UR-DNA regions within the UFBs require processing by the BTRR complex and topoisomerase 2 (TOP2). The resulting ssDNA regions are protected in so-called 53BP1 nuclear bodies in G1 and S phase until the next round of DNA replication [99,100,101]. These bodies serve to protect and mark DNA lesions and also provide the cell with a second opportunity to replicate loci with inherited UR-DNA in the following S phase. RIF1 and the shieldin complex direct the repair of these loci and restrict their replication to RAD52-mediated repair during the late S phase [102]. Whether such a second chance is a good option from the perspective of cells’ genomic stability is currently under discussion. First, the formation of 53BP1 nuclear bodies carries DNA damage to the next cell cycle with no guarantee of DNA repair, especially under conditions of additional replication stress. Second, this strategy implies that replication needs to “catch up” in the next round. Otherwise, those DNA regions would always be a cell cycle behind compared to the rest of the genome. Third, the involvement of RAD52 in the synthesis of UR-DNA at 53BP1 nuclear bodies increases the chances of DNA duplication by break-induced replication-like DNA synthesis, which is documented to be very mutagenic [103]. 

## 7. CIN-Dependent and Independent Cell Death 

As discussed in previous sections, cells have multiple mechanisms to ensure the proper replication and segregation of the genome into daughter cells. However, in cases where aberrations are not dealt with, they will accumulate, paving the way for cellular transformation. CIN is a distinctive feature of solid tumors and the main culprit behind common tumor phenotypes, such as aneuploidy and intra-tumoral heterogeneity. These phenotypes directly correlate with cellular transformation, tumor progression and recurrence, resistance to chemotherapy, and poor prognosis [104,105,106,107,108]. On the other hand, the accumulation of CIN can also lead to cell death, which is a desirable scenario in the context of tumor cells (Figure 2). 

The double role that CIN plays in cancer sets the stage for two opposite strategies during cancer treatment. One strategy is the very challenging choice of attempting CIN reduction without affecting the treatment’s ability to induce cell death. This option would reduce the overall rate of genetic variation and intra-tumor heterogeneity, thus impacting tumor characteristics such as adaptability, metastasis, and drug resistance. An opposite strategy is to exacerbate CIN to a level that can only drive cell death. To this end, in the last years, there have been an increasing number of studies that show that CIN can sensitize cells to specific treatments and be used as an Achilles’ heel to kill cancer cells. Some of these strategies are briefly discussed below.

### 7.1. CIN-Induced Cell Death in M Phase

One of the best examples of an approach in which pushing the limits of CIN induces cell death is the synthetic lethality observed when cells deficient in Breast Cancer Susceptibility Proteins, BRCA1 and BRCA2, are treated with poly (ADP-ribose) polymerase (PARP) inhibitors [109,110,111]. PARP inhibitors (PARPi) trap PARP on the DNA, creating roadblocks for replication and thus increasing the need for DNA repair mechanisms that deal with replicative DNA damage, such as HR [112]. Because BRCA cells already lack functional HR and display many CIN markers, the PARPi-induced lesions create a surplus in the load of DNA damage that cells can no longer handle. Interestingly, PARPi-mediated cell death in BRCA-deficient cells requires progression through mitosis, which gives rise to chromosome bridges derived from the high levels of replicative stress [113]. 

The above example suggests that a valid strategy to induce cell death is to generate significant levels of mitotic defects and drive cells to death by mitotic catastrophe [114]. Such a scenario can be propitiated by exacerbating replication defects in S or altering M phase regulation or its length. An example of such a strategy includes the downregulation of Aurora kinases, which alters the function of critical mitotic components such as centrosomes and microtubule-kinetochore interactions, and increases chromosome missegregation and multinucleation, ultimately driving cell death in certain types of tumors such as breast, thyroid, and colon [115,116,117,118,119,120]. Similar phenotypes were obtained when S-phase defects were exacerbated. For example, BRCA2-deficient cells exhibit UR-DNA features, and their survival depends heavily on the expression of MUS81, an endonuclease required for MiDAS. In the absence of MUS81 activity, BRCA2 cells display increased CIN phenotypes such as anaphase bridges, whose lack of resolution seemingly leads to multinucleation and cell death [121]. An intriguing commonality among these studies is the generation of multinucleated cells through the induction of mitotic defects. In this sense, multinucleation induction may function as a tumor suppressor phenotype that could potentiate CIN-inducing therapies (Figure 2). Lastly, MN induction can also tilt the balance toward apoptosis [93]. Although MN can derive from multiple origins (Figure 1), their formation requires passage through mitosis (Figure 2), suggesting that inducing mitotic defects that lead to their expression could also be used to trigger M phase-related cell death. Ultimately, CIN-mediated genomic instability can only be sustained as long as the cell does not reach an upper threshold where it will inevitably trigger mitotic cell death mechanisms [117,122]. Even though mitotic kinase inhibition can be used to push cells towards this threshold, limitations such as off-target effects, toxicity to normal cells, and induction of secondary tumors, among others, need to be considered.

### 7.2. CIN-Independent Cell Death in S Phase

An alternative strategy to cell death by mitotic catastrophe is to induce replication catastrophe and force cell death during the S phase before cells reach mitosis, thus avoiding the potential risk of genetically unstable daughter cells [123,124]. Replication catastrophe refers to a massive disruption of DNA replication, such as that caused by replication fork collapse across the entire genome. One such example of a replication catastrophe is observed with ATR inhibition, whose massive effect on replication forks leads to the exhaustion of RPA. This protein coats ssDNA on stalled or collapsed forks and serves as a trigger for DNA repair [125]. When ATR is inhibited, downstream effectors such as CHK1 cannot rescue stalled replication forks augmenting ssDNA regions and the demand of ssDNA coating factor, RPA. Usually, RPA levels exist in excess when compared to the levels of ssDNA. However, under the above conditions, this is no longer the case, and RPA exhaustion leads to massive amounts of unprotected ssDNA, which can break, causing a G2/M arrest, senescence, and death by replication catastrophe [125]. Interestingly, RPA exhaustion may not only be triggered by checkpoint regulation. The downregulation of translesion synthesis polymerase η, which is overexpressed in cisplatin-resistant tumors, also triggers a type of cell death that happens in S phase, is regulated by RPA exhaustion, and it does not augment CIN [126]. Such results suggest that different DDR effectors could be regulated to tilt the balance toward the induction of cell death in S phase, in a manner that precludes the generation of CIN.

## 8. CIN and Cancer Therapies

Radiation and chemotherapy are the staples of traditional cancer therapy approaches. Mechanistically, their rationale is based on exploiting the faster division of cancer cells compared to healthy cells. As such, these approaches challenge DNA replication (e.g., radiation, platinum compounds) or mitotic processes such as microtubule dynamics (e.g., paclitaxel) and induce cytotoxic cell death triggered by excessive CIN [127,128,129]. Although their success cannot be understated, these therapies are non-selective, generate resistant tumor populations, and target rapidly dividing healthy cells such as those found in the intestinal epithelium, hair follicles, and bone marrow, among others [130,131]. The death of the non-tumor is responsible for the significant side effects associated with chemotherapy, such as hair loss, pain, nausea, diarrhea, cardiotoxicity, and immune suppression [132,133]. Because these therapies are widely reviewed in the literature, in this section, we will focus on targeted therapies that take advantage of different aspects of CIN.

As can be inferred from its name, targeted therapy aims to kill tumor cells by disrupting the specific molecular players that allow their growth, progression, spread, and survival [134]. Approaches for targeted therapy are diverse and include hormone therapies, signal transduction inhibitors, angiogenesis inhibitors, apoptosis inducers, monoclonal antibodies that deliver toxic molecules, gene expression modulators, and immunotherapy [134]. Although it was initially expected that side effects and resistance could be mostly avoided when choosing these strategies, targeted therapies also face their own challenges. 

Decades of cancer research have shown that a central issue is the emergence of residual populations that develop resistance to treatments that were initially effective. Resistance mechanisms may vary (e.g., alterations of the pumps that control the entrance of chemicals into cells or the specific pathways targeted by such chemicals), but they are all triggered by the mutagenesis capacity of cancer cells. Below we will discuss the strategies aimed to push mutagenesis to a level that guarantees cell killing with no survival of the mutagenized population (Figure 3). Alternatives such as keeping the mutagenesis burden in check or using acute CIN as a trigger to facilitate immunotherapy will also be discussed below.

### 8.1. Increased CIN as a Tool to Kill Cancer Cells

Synthetic lethality is a term coined from developmental biology in which the combined deficiency of two gene products leads to cell death, while a single deficiency does not. In the context of cancer, one gene deficiency is intrinsic to tumor cells and is a consequence of a genetic deletion or mutation, while the second deficiency is achieved via pharmacological inhibition using small molecule inhibitors. Because non-tumor cells lack the genetic changes present in the tumor cells, synthetic lethality therapy approaches are intrinsically designed to only target tumor cells. 

The best example of a synthetic lethal approach is the use of PARPi in tumors with defective homologous recombination proteins BRCA1 and BRCA2 [109,110]. PARPi are among the most successful examples of precision medicine approaches and were also the first DDR inhibitors approved for clinical use by the U.S. Food and Drug Administration (FDA). The mechanisms for the synthetic lethality between PARPi and BRCA proteins have been reviewed elsewhere [135] but as mentioned above (Section 7.1.), it relies on the generation of DNA replication barriers, due to PARP trapping [112], that induce DSBs from collapsed forks that require HR for their resolution. Additionally, cell death by PARPi requires progression through mitosis, highlighting the connection between replication defects and proper mitosis [113,136]. Just like with classical chemotherapy, tumor resistance has also been observed with PARPi [135]. The recovery of the BRCA function has been reported to be the primary mechanism observed in the clinic. To circumvent resistance, therapies that include PARPi as an adjuvant to conventional chemotherapies have been tested and have sometimes cast promising results [129,137]. PARPi have also been combined with inhibitors for cell cycle checkpoint kinases such as ATR, CHK1, and WEE1 [138,139]; MUS81 nuclease inhibitors [140]; or signaling molecule inhibitors such as PI3K, AKT, or mTOR [141]. Inhibitors of alternative end-joining (Alt-EJ) polymerase theta are also under development as that appears to be the predominant repair choice for one-ended DSBs in HR-deficient cells [142].

In the context of PARPi, another therapeutic strategy being considered is to generate death in S phase through poly (ADP-ribose) glycohydrolase (PARG) inhibitors. PARG is the primary enzyme responsible for the catabolism of PARP-derived PARylation and, as such, has critical roles in maintaining the stability of replication forks under conditions of replication stress [143,144]. However, unlike PARPi, whose mechanism of death is via mitotic catastrophe [136], PARGi treatment induces an S/G2 arrest, and ssDNA accumulation and RPA depletion followed by an increase of DSBs, ultimately leading to death by replication catastrophe [145,146]. More importantly, it has also been observed that PARGi can kill PARPi-resistant breast and ovarian tumor cells, suggesting that co-treatment with PARPi and PARGi could be beneficial [147]. 

Besides PARPi, other small molecule inhibitors that target apical kinases of the DDR or mitosis, such as ATR, ATM, PLK1, and AKT, which may target HR-deficient tumors either alone or in combination with chemotherapies, are already in clinical trials for their use in cancer treatments [148,149,150]. However, more research is needed to overcome limitations to small molecules, such as resistance, low specificity, and short lifespan.

### 8.2. CIN-Independent Cell Death

Because CIN is considered a catalyzer of tumor adaptation to treatment, a very challenging and under-explored option is to design protocols that induce cell death without generating abrupt and acute changes in CIN levels. In this regard, a recent report shows that BRCA1-deficient cells are killed when treated with the PLK1 inhibitor volasertib [150]. Interestingly, this cell death does not require DNA damaging agents and selectively affects the BRCA1-deficient background with a much more modest effect in BRCA2-deficient cells. These observations, along with the fact that PLK1 is a mitotic kinase, indicate that cell death is triggered outside S phase. Moreover, while PARP inhibition causes acute accumulation of replication stress and CIN markers, PLK1 inhibition does not. Whereas the mechanism of BRCA1-deficient cell killing by PLK1 is still obscure, a retrospective analysis reveals that BRCA1-deficient human tumors may frequently become addicted to PLK1 [150]. Acute CIN could be prevented by employing novel treatments or by adapting treatments involving agents that induce acute CIN. For example, CHK1 inhibitors induce cell death and accumulation of both bulky chromosome bridges and MN. A mitosis-specific pathway involving MUS81 endonuclease, its partner EME1, and MiDAS components RAD52 and POLD3 promote the accumulation of such CIN markers, with no effect on the extent of cell death [72]. While it is still unclear whether precluding acute CIN during cancer treatments suffices to prevent the acquisition of genome instability, it is tempting to speculate that the limitation of acute CIN could benefit the outcome of cancer treatments.

### 8.3. CIN as a Facilitator of Immunotherapy

Immunotherapy is a targeted therapy that boosts or modifies the immune system to improve its capacity to detect and kill cancer cells. Immunotherapy includes the use of monoclonal antibodies, cancer vaccines, immunomodulators, cytokines, CAR T-cell therapies, and immune checkpoint blockers [151]. For this review, we will briefly discuss some recent connections between immune checkpoint blockades and CIN. 

Immune checkpoint inhibitors such as anti-Programmed Cell Death Protein (PD1), anti-Programmed Death-Ligand 1 (PD-L1), and anti-Cytotoxic T-Lymphocyte Protein 4 (CTLA-4) have recently been approved for cancer treatment use due to their potent antitumor effects. These inhibitors release the breaks of the immune system by inhibiting signals that are otherwise inhibitory of T cell responses, thus reducing the threshold for immune recognition of the tumor [152]. Currently, immune checkpoint blockers are being used as neoadjuvant or adjuvants to other therapies, including PARPi [153,154], which has been shown to modulate the tumor immune microenvironment through the generation of damaged DNA and via its PARP-trapping activities [155,156,157,158]. MN and bulky chromosome bridges can give rise to cytosolic DNA, which can activate the cyclic GMP-AMP synthase (cGAS)-stimulator of interferon genes (STING) pathway, a pathogenic and cytoplasmic DNA surveillance pathway. c-GAS binds double-stranded DNA in the cytosol, triggering cytokine production, and promoting T cell immune response against tumors [159,160,161,162].

Additionally, this response can also increase the expression of PD-L1, which could potentially make tumors more sensitive to PD-L1 blockers [159,160]. Interestingly, cytosolic DNA derived from ruptured MN is also subject to attack by the TREX1 nuclease, which can prevent their immunomodulatory effects, and c-GAS has been shown to reduce MN abundance via autophagy-mediated clearance, suggesting a tight regulation of the cGAS-STING pathway [6,7]. c-GAS-STING can also sense mitotic chromosomes, which lack a nuclear envelope, and promote cell death upon prolonged mitotic arrest triggered by a wide array of DNA damage besides MN [163]. Altogether, the research indicates that CIN is not only sensed at multiple levels but can be tightly regulated, and these novel intricate mechanisms could be modulated to potentiate immunotherapy (Figure 4). It has also been reported that other molecules that target the DDR and related pathways have shown additive or synergistic effects when combined with immune checkpoint blockers. Among these are alisertib, an aurora A kinase inhibitor [164], PLK1 inhibitors [165], the ATR inhibitor AZD6738 [166], and the CHK1 inhibitor SRA737 [167]. 

Radiation therapy, which also induces CIN and CIN-mediated cell death, has also shown promising results in combination with immunotherapies. As a localized therapy, radiation-induced cell death activates the immune system locally, inducing neoantigen expression and increasing antigen-presenting cells capable of exhibiting tumor neoantigens on their surface. Additionally, radiation promotes the release of pro-inflammatory cytokines, which serve to increase tumor T-cell infiltration. Given that these immunostimulating events can be counteracted by PD-L1 expression in tumor cells, it is not surprising that inhibiting PD-L1 by immunotherapy approaches such as PD1 antibodies (i.e., pembrolizumab) shows favorable results in combination with radiation [168,169]. 

Given the advancements in immunotherapy, understanding the mechanisms by which DNA damage regulates the immune responses will be crucial for developing new therapies. Such research will be of particular relevance in designing treatments for solid tumors in which CIN undoubtedly plays a role in tumor progression and resistance. 

## 9. Concluding Remarks

As discussed above, structural CIN is a common feature of solid tumors, and the last decades have yielded a great understanding of the molecular mechanisms that give rise to these complex aberrations. It is currently accepted that structural CIN parameters have outstanding importance in the clinic. Not only is CIN a hallmark of tumors, but it is also a tool for prognosis. More importantly, it can be exploited to better inform treatment options. However, information relevant to defining which CIN markers are better indicators of tumor prognosis and response is limited. The usefulness of micronuclei to distinguish malignant lesions from benign lesions using cytological specimens is well established [170]. Interestingly, many of the structural CIN aberrations can eventually lead to micronuclei formation (Figure 1), and additionally, micronuclei have been shown to trigger apoptosis and mediate immune responses (Figure 4). This poses the question of whether this specific type of aberration could serve as a staple marker to analyze in CIN research studies and predict long-term treatment outcomes. In the same direction, the use of structural CIN profiles as predictors of thresholds that promote cell death but not tumor aggressiveness or heterogeneity would be of utmost importance for the design of precision medicine approaches (Figure 3).

From the perspective of understanding CIN in the experimental laboratories, live-cell microscopy has been the method of choice when attempting to understand both the origin and dynamics of different aberrations. Moreover, cell recording is the ultimate and only tool that unequivocally reveals the fate of cells that present CIN. 3-D organoid tissue culture, such as tumor patient-derived organoids, is the model that most resembles human tumors in tissue cultures, and is a crucial tool that must be used for high temporal resolution analysis of cellular events, including CIN. Complementing such technologies with other parameters such as histology, single-cell DNA and RNA sequencing, OMICS approaches, and drug screenings will shed rapid light on the advantages/disadvantages resulting from CIN accumulation after each treatment. 

In conclusion, while more research is still needed to fully profit from structural CIN studies in clinical settings, progress has been outstanding in the last decade. It has long been known that cancer cells are enriched in altered chromosomes, but only recently have researchers managed to establish an association between these gross DNA alterations and the molecular pathways that have triggered them. Such discoveries have paved the way to use such markers to design more effective and precise cancer treatments, an opportunity that is just beginning to be exploited and from which cancer treatments are expected to benefit significantly.

## Figures and Tables

**Figure 1 cancers-13-03056-f001:**
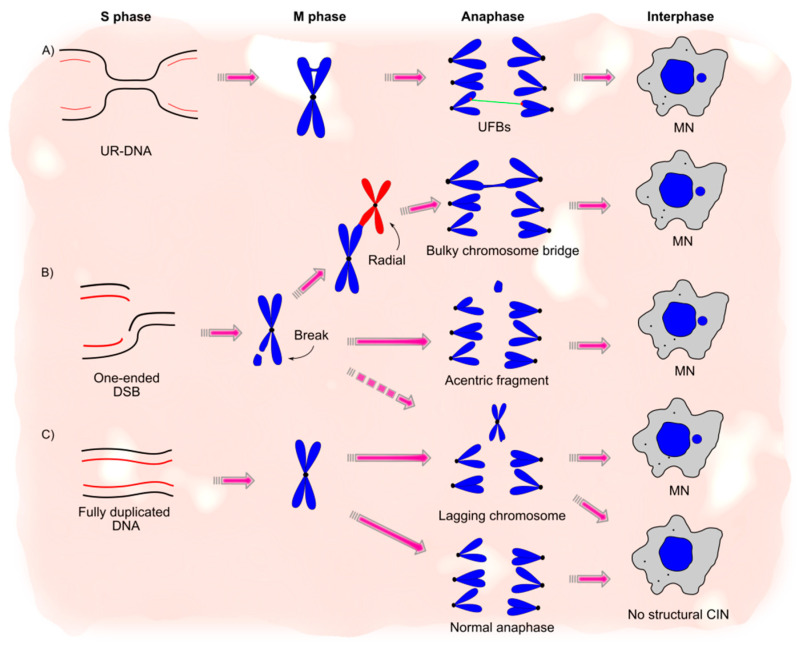
The multiple origins and tight associations of chromosome instability (CIN) phenotypes. (**A**) Under-replicated DNA (UR-DNA) formed during S phase and not resolved before mitosis leads to ultra-fine bridges (UFBs) when sister chromatids are pulled in opposite directions during anaphase. Breakage of these bridges can eventually lead to micronuclei (MN) formation in the following cell cycle. (**B**) Unrepaired one-ended double-strand breaks (DSBs) formed during S or M phase due to replication fork collapse can be visualized as chromosome breaks during mitosis. Improper repair of breaks by end-joining mechanisms can create fusions among non-homologous chromosomes, giving rise to radial chromosomes. Radial chromosomes form bulky chromosome bridges due to multiple centromeres and unequal pulling toward opposite poles. Similarly to UFBs, breakage of these bridges can also lead to MN formation. Breaks that are not fixed and lack a centromere give rise to acentric fragments, while breaks that activate the DNA damage response and DNA repair during mitosis (dotted arrow) can form lagging chromosomes. Both acentric fragments and lagging chromosomes are well known sources of MN. (**C**) Fully duplicated chromosomes mostly lead to normal cell division in mitosis and absence of structural CIN. However, under certain circumstances, they can experience lagging at the metaphase plate during anaphase, usually due to kinetochore-microtubule attachment problems. These lagging chromosomes can form whole chromosome micronuclei in the next cell cycle.

**Figure 2 cancers-13-03056-f002:**
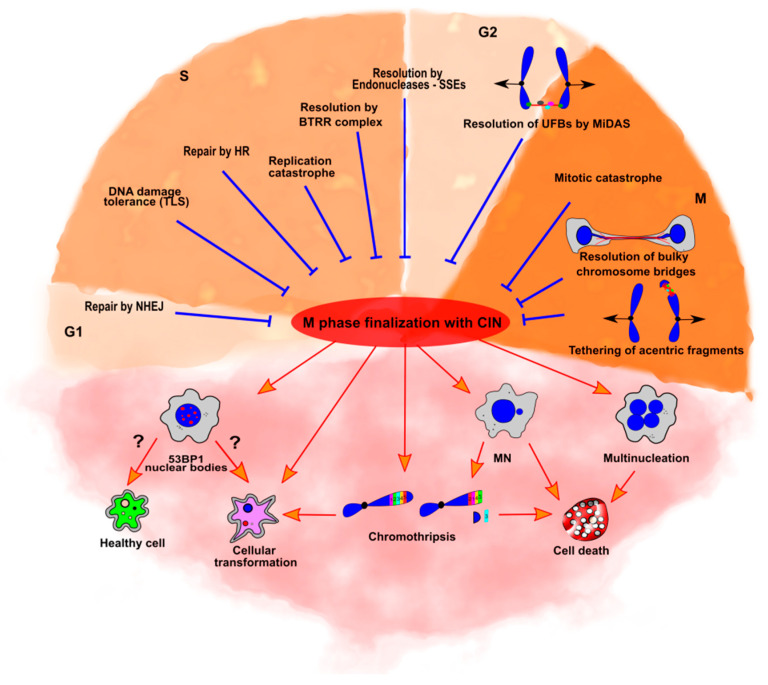
Mechanisms that inhibit or promote chromosome instability (CIN). Top (blue lines): Mechanisms involved in inhibiting or restraining CIN during the different cell cycle stages. Non-homologous end joining (NHEJ) can be utilized throughout the cell cycle but is required for double-strand break (DSB) repair in G1 phase due to the absence of homologous recombination (HR). Translesion synthesis (TLS), HR, and multiple endonuclease-dependent mechanisms, such as the BTRR complex and structure-specific endonucleases (SSEs), can prevent the accumulation of CIN during mitosis. DNA bridge resolution and mitotic DNA synthesis (MiDAS) during M phase can resolve multiple types of DNA bridges. Broken chromosome fragments can be incorporated in M phase via chromosome tethering mechanisms. Replication or mitotic catastrophe in S or M phase, respectively, when exacerbated, usually leads to cell death, thus preventing the accumulation of cells with CIN. Bottom (orange arrows): Cells that enter mitosis with chromosome abnormalities due to failed repair mechanisms present alternative mechanisms of resolution that can potentiate CIN and carcinogenesis, such as micronuclei (MN) breakage via chromothripsis. Some forms of DNA damage lead to 53BP1 nuclear bodies, whose implications are still largely unknown. In contrast, some MN or gross mitotic abnormalities, such as multinucleation, can lead to cell death.

**Figure 3 cancers-13-03056-f003:**
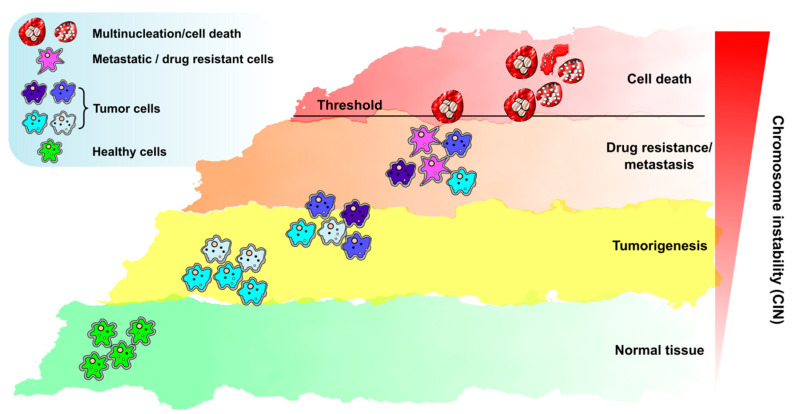
The multifaceted role of CIN in carcinogenesis. Normal tissue (bottom, green shade) can become tumorigenic under circumstances directly or indirectly associated with the induction of replication stress (yellow shade). At that point, tumor cells commonly present a low level of chromosome instability (CIN), which can serve as a fuel for early steps of cellular transformation and thus enable the carcinogenic process (yellow shade). As the tumor progresses (orange shade), cells become increasingly genetically unstable. Under these conditions, CIN can promote the fast acquisition of multiple tumor characteristics such as drug resistance and metastasis. Eventually, if CIN levels are very high, cells can reach a threshold in which they are no longer viable (red shade). Surviving cells in these stages are highly genetically unstable, with an increased probability of being multinucleated after subsequent cycles of aberrant mitosis finalization. Such a scenario has good chances of triggering cell death due either to improper gene expression or to sub-optimal S and M phase finalization.

**Figure 4 cancers-13-03056-f004:**
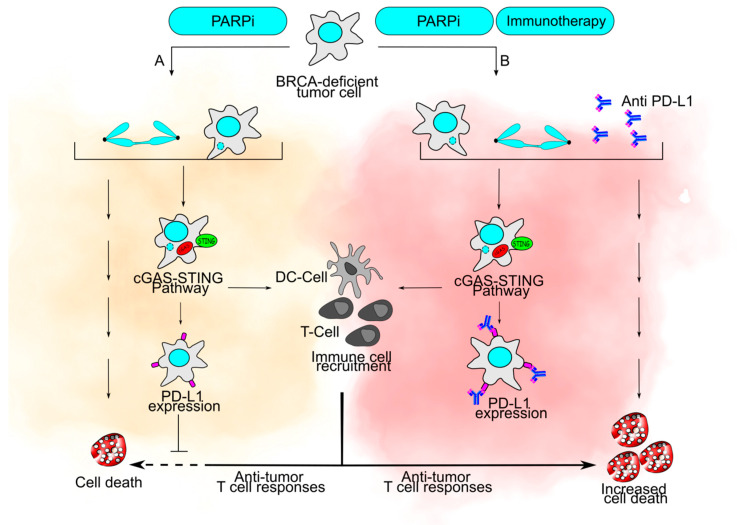
The potent combination of chemotherapy and immunotherapy. (**A**) Breast Cancer Susceptibility Protein (BRCA)-deficient tumor cells treated with Poly(ADP ribose) Polymerase inhibitor (PARPi) exhibit under-replicated DNA and unrepaired double strand breaks in S phase, which trigger bulky chromosome bridges and micronuclei formation, eventually causing cell death. Micronuclei also trigger the cyclic GMP-AMP synthase-stimulator of interferon genes (cGAS-STING) immune pathway, which activates dendritic (DC) and T cells, which are recruited to the tumor. However, their anti-tumor activity is blocked due to Programmed Death-Ligand 1 (PD-L1) expression. (**B**) PARPi treatment combined with immunotherapy such as anti-PD-L1 antibodies also leads to bulky chromosome bridges, micronuclei and cGAS-STING pathway activation. However, under these conditions, the PD-L1 antibodies can block PD-L1, leading to an increased anti-tumor cell response mediated by T-cells.

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
