# Peer review of "Structural Chromosome Instability: Types, Origins, Consequences, and Therapeutic Opportunities"

_cancers, 2021, doi:10.3390/cancers13123056_

Round 1

Reviewer 1 Report

This is a very interesting, well written and thorough review that addresses all the key aspects of chromosome instability. It is structured well and is easy to understand. I think this will be a very valuable addition to the field. The only issues I identified were very minor: some typos and the excessive use of acronyms that makes it difficult for one to follow. 

Minor typos:

  1. Line 121: change bridge's to bridged
  2. Line 245: change occasions to occasion
  3. Line 301: the work deriving seems out of place
  4. Please reduce the use of acronyms. The commonly used ones are pretty easy to remember but the new acronyms are really hard to keep track of. I had to keep referring to the earlier sections.
  5. In Figure 2: the spelling of "replicaction catastrophe" needs to be changed
  6. Figure 4: spelling of "PD-L1 espression" needs to be changed.
  7. Line 551: change “preponderant repair choice” to predominant repair choice

Reviewer 2 Report

Siri et al. summarized the latest research aspects in structural CIN generation and therapeutic potentials. The review was clearly written with sufficient citations. Some suggestions to improve the manuscript:

  1. Regarding the origin of CIN, the formation of MN was emphasized and listed nearly exclusively. While it is acceptable to focus just on MN generation, this should be reflected in the Abstract. A summary description at the beginning of Section “3. Cellular phenotypes associated with CIN” stating the emphasis on MN should also be included. Title of Figure 1 should be altered accordingly with not so broad definition.
  2. Spell out of the abbreviations in all figure legends.
  3. Resolution of Figure 2 is too low. Resolution of other figures could also be improved.
  4. “PD-L1 espression” should be “PD-L1 expression” in Figure 4.
  5. Line 542, “thru” should be through; Line 550, is the symbol after “polymerase” supposed to be Theta?
